# Anti-Inflammatory Effects of Aurantio-Obtusin from Seed of *Cassia obtusifolia* L. through Modulation of the NF-κB Pathway

**DOI:** 10.3390/molecules23123093

**Published:** 2018-11-27

**Authors:** Jingyi Hou, Yu Gu, Shuai Zhao, Mengqi Huo, Shifeng Wang, Yanling Zhang, Yanjiang Qiao, Xi Li

**Affiliations:** 1Research Center of Traditional Chinese Medicine Information Engineering, School of Chinese Materia Medica, Beijing University of Chinese Medicine, Beijing 100102, China; hjy_2016@126.com (J.H); riberyguyu@163.com (Y.G.); 20170931863@bucm.edu.cn (S.Z.); 20150931715@bucm.edu.cn (M.H.); alana6268@126.com(S.W.); xixili1994@163.com (X.L.); 2Beijing Key Laboratory of Traditional Chinese Medicine Basics and New Drug Research, Research Center of Traditional Chinese Medicine Information Engineering, Beijing University of Chinese Medicine, Beijing 100102, China

**Keywords:** inflammation, NF-κB, *Semen Cassiae*, anthraquinone

## Abstract

Aurantio-obtusin, an anthraquinone compound, isolated from dried seeds of *Cassia obtusifolia* L. (syn. *Senna obtusifolia*; Fabaceae) and *Cassia tora* L. (syn. *Senna tora*). Although the biological activities of *Semen Cassiae* have been reported, the anti-inflammatory mechanism of aurantio-obtusin, its main compound, on RAW264.7 cells, remained unknown. We investigated the anti-inflammatory effect of aurantio-obtusin on lipopolysaccharide- (LPS)-induced RAW264.7 cells in vitro and elucidated the possible underlying molecular mechanisms. Nitric oxide production (NO) and prostaglandin E_2_ (PGE_2_) were measured by the Griess colorimetric method and enzyme-linked immunosorbent assay (ELISA), respectively. Protein expression levels of cyclooxygenase 2 (COX-2) were monitored by cell-based ELISA. Interleukin 6 (IL-6) and tumor necrosis factor-alpha (TNF-α) synthesis were analyzed using ELISA. The mRNA expression of nitric oxide synthase (iNOS), COX-2, and the critical pro-inflammatory cytokines (IL-6 and TNF-α) were detected by quantitative real-time PCR. Aurantio-obtusin significantly decreased the production of NO, PGE_2_, and inhibited the protein expression of COX-2, TNF-α and IL-6, which were similar to those gene expression of iNOS, COX-2, TNF-α and IL-6 (*p* < 0.01). Consistent with the pro-inflammatory gene expression, the Aurantio-obtusin efficiently reduced the LPS-induced activation of nuclear factor-κB in RAW264.7 cells. These results suggested that aurantio-obtusin may function as a therapeutic agent and can be considered in the further development of treatments for a variety of inflammatory diseases. Further studies may provide scientific evidence for the use of aurantio-obstusin as a new therapeutic agent for inflammation-related diseases.

## 1. Introduction

Inflammation is essential in the complex network of immunological and physiological defense processes which occur following noxious stimulation. The stimulations include infection, toxin exposure, tissue injury, or exposure to endotoxins [1,2,3,4]. Unwanted and prolonged inflammatory responses are harmful to the body and function as a potential risk and pivotal drivers, in a number of major diseases. Cancer [5], diabetes [6], rheumatoid arthritis [7], chronic inflammatory bowel disease [8], cardiovascular disease [9], multiple sclerosis [10], and psoriasis [11] are some of the major diseases that may be affected by these responses. One of the key events involved in the inflammatory process includes the expression of inflammatory cytokines, chemokines, the signaling pathway, and other mediators (inducible NOS) [12].

Macrophages, a well-known inflammatory and immune effector cell type, play crucial roles in the immune response, in inflammatory diseases, and in host defense mechanisms, through the production of pro-inflammatory cytokines and inflammatory mediators [13]. Lipopolysaccharide (LPS), one of the most powerful activators of macrophages, activate these cells via the release of inflammatory mediators, cytokines and chemotactic factors. Toll-like receptor 4 (TLR4), one of the LPS recognition receptor, which triggers downstream of nuclear factor kappaB (NF-κB) [14]. NF-κB, a five-membered collection of transcription factors, is involved in physiological and patho- physiological responses [15]. In addition, this protein regulates the expression of genes and cytokines which play crucial roles in apotheosis, inflammation, proliferation, and calcium homeostasis [16,17,18]. NF-κB p50/p65, two subunits of NF-κB [19], once encountered inflammatory stimuli such as LPS, which increase the nuclear p65 protein, decreases cytosolic I-κB protein, promotes the expression of its downstream pro-inflammatory mediators and cytokines (i.e., inducible NOS, cyclooxygenase-2 (COX-2), moocyte chemoattractant protein 1(MCP-1), IL-6, IL-1β, tumor necrosis factor—α (TNF-α) and others [19,20]), and eventually induces an inflammatory response [21]. Therefore, the NF-κB pathway is emerging as potential therapeutic targets for the remedy the inflammatory diseases [22].

Over the past century, natural products, especially anthraquinone compounds, have become valuable products for achieving chemical diversity in the molecules used for inflammation relief [23,24]. 1,3,7-Trihydroxy-2,8-dimethoxy-6-methyl-9,10-anthracenedione (aurantio-obtusin, Figure 1), is a reported anthraquinone derivative, and is the major bioactive compound obtained from the dried seeds of *Cassia obtusifolia* L. (syn. *Senna obtusifolia*; Fabaceae) and *Cassia tora* L. (syn. *Senna tora*) [25,26,27] as well as a phytochemical marker of quality control in the Chinese Pharmacopeia (Version 2015) [28].

Accumulated evidences have shown that aurantio-obstusin exhibits a number of biological properties including anti-oxidative, anti-hypertension, anti-mutagenic, anti-genotoxic, anti-allergic and neuroprotective effects [3,29,30,31,32]. *Semen Cassiae* is widely used in Chinese folk medicine and has been demonstrated to exhibit significant anti-inflammatory effects [33,34]. Whether aurantio-obtusin plays a pivotal role in the anti-inflammatory effects exhibited by *Semen Cassiae*, remains unclear. Likewise, the underlying intracellular mechanisms whereby aurantio-obtusin exhibit its anti-inflammatory effect require further investigation. In this study, LPS-stimulated RAW264.7 cells were used as an inflammatory model to investigate the anti-inflammatory potency of the compound, aurantio-obtusin, isolated from *Semen Cassiae*. In addition, we investigated the underlying mechanism involved in the anti-inflammatory effects exhibited by this compound. Several chemokines, cytokines, and enzymes involved in the inflammatory process including NO, IL-6, iNOS, TNF-α, PGE_2_ and COX-2, were investigated to demonstrate these anti-inflammatory effects. We also investigated whether aurantio-obtusin exerts its anti-inflammatory activity by modulating the NF-κB pathway.

## 2. Results

### 2.1. Effect of Aurantio-Obtusin on Cell Cytotoxicity

MTT cell viability assays were used to measure the cytotoxicity of aurantio-obtusin in RAW264.7 cells. As shown in Figure 2, the cell viability of RAW264.7 was not significantly affected by aurantio-obtusin at concentrations up to 100 μM, while concentrations below 100 μM did not affect the cells (*p* < 0.05). As a result, the RAW264.7 cells were treated with aurantio-obtusin concentrations ranging from 6.25 to 50 μM for all subsequent experiments.

### 2.2. Effect of Aurantio-Obtusin on NO Production in RAW264.7 Cells

The effect of aurantio-obtusin on NO production in LPS-treated RAW264.7 cells were measured to investigate its anti-inflammatory effects. The concentrations of nitrite accumulated in the culture medium was tested using Griess reagent to serve as an index for NO.

The RAW264.7 cells were pretreated with different concentrations of aurantio-obtusin (6.25–50 μM) for 2 h, and stimulated with 0.2 μg/mL LPS (Figure 3A). Furthermore, laser scanning microscopy also shown aurantio-obtusin to be a stronger inhibitor of intracellular NO production than that in single LPS stimulation (Figure 3B). Dexamethasone, a NO inhibitor, used as a positive control, also inhibited the production of NO in the activated RAW264.7 cells (Figure 3).

### 2.3. Effect of Aurantio-Obtusin on the LPS-Induced Cytokines, TNF-α, IL-6, COX-2 and PEG_2_

Pro-inflammatory cytokines, such as TNF-α and IL-6, have been shown to pose a multitude of biological effects linked to autoimmunity, and acute or chronic inflammatory diseases. Thus, we evaluated the inhibitory effects of aurantio-obtusin (6.25–50 μM) on pro-inflammatory cytokines in LPS-induced RAW264.7 cells. As shown in Figure 4A,B, pretreatment of RAW264.7 cells with aurantio-obtusin (25, 50 μM) significantly suppressed the IL-6 production, while TNF-α was only significantly reduced when the aurantio-obtusin concentration was 25 μM (*p* < 0.01).

PGE_2_ is synthesized by the catalytic conversion of arachidonic acid to prostaglandin H_2_ by COX-2, PGE_2_ function as another import mediator in the pathogenesis of inflammation. To investigate the inhibitory activities of aurantio-obtusin on PGE_2_ and the protein expression of COX-2, RAW264.7 cells were cultured in a 96-well plate treated with the indicated concentrations of aurantio-obtusin (6.25–50 μM) and stimulated with 0.2 μg/mL LPS for 24 h. As shown in Figure 4C, LPS treatment significantly increased PGE_2_ production, while 50 μM aurantio-obtusin exhibited a significant inhibitory effect on PGE_2_ production (*p* < 0.01).

To examine whether the suppression of PGE_2_ can be attributed to the decreased protein expression of COX-2, the effect of aurantio-obtusin (6.25–50 μM) on COX-2 protein expression was determined. As shown in Figure 4D, the LPS-stimulated high expression of COX-2 was significantly inhibited in cells pretreated with aurantio-obtusin in a concentration-dependent manner. In addition, only 50 μM concentration of aurantio-obtusin significantly inhibited the protein expression of COX-2 (*p* < 0.01).

### 2.4. Effect of Aurantio-Obtusin on LPS-Induced mRNA Expression of TNF-α, IL-6, iNOS and COX-2 in LPS-Stimulated RAW264.7 Cells

IL-6, TNF-α, and pro-inflammatory mediators such as NO and PGE_2_, produced in the cell supernatant, were significantly inhibited (after LPS stimulation) by the addition of aurantio-obtusin in a concentration-dependent manner. To investigate the effects of aurantio-obtusin on LPS-induced mRNA expression, real-time PCR was performed. The inhibitory effects of aurantio-obtusin on LPS-induced mRNA expression of IL-6, iNOS, COX-2, and TNF-α were observed (Figure 5). The results suggested that the effect of aurantio-obtusin on mRNA expression of IL-6, TNF-α, iNOS, and COX-2 (*p* < 0.01) using all tested concentrations; a 56.41%, 54.54%, 92.02% and 76.95% significant inhibition were observed for IL-6, TNF-α, iNOS, and COX-2, respectively. In addition, the mRNA expression of IL-6, TNF-α, iNOS, COX-2 coincided with the secretion of IL-6, TNF-α, NO, PEG_2_ and COX-2 in the culture medium.

### 2.5. Effects of Aurantio-Obtusin on Proteins, with the Inhibition of NF-κB in LPS-Stimulated RAW264.7 Mouse Macrophages

To further confirm the anti-inflammatory properties of aurantio-obtusin, we evaluated its effect on the phosphorylation of inhibitor of NF-κB (IκB), and the protein expression of NF-κB p65, using a western blot assay. These components were selected for evaluation as they are involved in the pathogenesis of inflammation-related diseases. The β-actin protein was used as the internal control. LPS markedly induced the phosphorylation of IκB when the RAW264.7 cells were exposed to 0.2 μg/mL LPS for 12 h. Pretreatment with aurantio-obtusin effectively suppressed these processes in a dose-dependent manner, and significantly inhibited the protein expression of NF-κB p65. These results suggest that aurantio-obtusin inhibited the NF-κB-dependent inflammation pathway in RAW264.7 cells (Figure 6). Inhibitor of nuclear factor kappa-B kinase (IKK) can phosphorylate IκB, by acting as the upstream kinase of IκB in the NF-κB signaling pathway. Thus, the effects of aurantio-obtusin on LPS-induced IKK-α, -β activation were observed; aurantio-obtusin (12.5, 25, 50 μM) inhibited the expressions of IKK-α and IKK-β. As shown in Figure 7, LPS treatment resulted in the translocation of NF-κB from the cytoplasm to the nucleus as indicated by the detection of the Cy3-conjugated anti-NF-κB antibody and the blue fluorescence of nuclear counterstain DAPI. In contrast, the translocation of NF-κB was reduced in aurantio-obtusin treated cells.

## 3. Discussion

In this study, aurantio-obstuin, isolated from *Semen Cassiae*, was used to investigate its effects on LPS-induced inflammatory responses in the murine macrophage cell line, RAW264.7, model. aurantio-obtusin was shown to prevent inflammation through an underlying mechanism involved in the activation of NF-κB in LPS-stimulated RAW264.7 cells.

Inflammation, a basic defense mechanism, serves a variety of physiological purposes that include host defenses, tissue repair response, and recovery of the homeostatic state [35]. LPS, a major inducer of the outer cell membrane of gram-negative bacteria, promotes the activation of monocytes and macrophages. This in turn promotes inflammatory pathogenesis via stimulating the TLR4 and releasing of many inflammatory cytokines and mediators such as TNF-α, IL-6, IL-1β, and IL-18 [36].

A considerable amount of attention has been directed at aurantio-obtusin, an anthraquinone isolated from *Semen Cassiae*, as a result of its anti-oxidative and neuroprotective properties [27]. Our study was, however, the first conducted to investigate the underlying effects of aurantio-obtusin on the production of pro-inflammatory cytokines and the activation of inflammation in RAW264.7 macrophages. The inflammatory model of RAW264.7 macrophages stimulated by LPS, is considered a canonical model for inflammation research. This is because this model can be used to investigate the effects of pharmacological treatment on the systemic inflammatory response. The experimental data demonstrated that aurantio-obtusin exhibited non-cytotoxic effects on RAW264.7 cells when combined with 0.2 μg/mL LPS stimulation.

NO, an intracellular messenger and regulator of inflammatory responses, was produced in high amounts by the up regulation of iNOS expression in activated inflammatory cells [37]. However, pretreatment with aurantio-obtusin was found to significantly suppress LPS-stimulated NO production in a concentration-dependent manner. In addition, the inhibitor, aurantio-obtusin (12.5–50 μM), was observed to exhibit stronger inhibition than dexamethasone (100 μM), the positive control. The mRNA expression of iNOS was also inhibited by aurantio-obtusin. This suggests that the suppressing effect of aurantio-obtusin on LPS-induced iNOS expression can be attributed to the transcriptional inhibition of the iNOS gene [38].

During the inflammation process, LPS has been reported to stimulate macrophage activation, resulting in the release of pro-inflammatory cytokines [39]. TNF-α and IL-6 are considered as critical cytokines involved in the inflammatory responses and their suppression can be regarded as a therapeutic strategy for inflammation-related disorders [40,41]. Hence, we selected TNF-α and IL-6 as the critical indicators of the inflammatory response, to investigate the anti-inflammatory effect of aurantio-obtusin. In the present study, pre-treatment with aurantio-obtusin suppressed TNF-α and IL-6 production and their mRNA expression. More importantly, the suppression of IL-6 production and its mRNA expression, were more significant than that of TNF-α when pretreated with aurantio-obtusin. This suggested the potential therapeutic effect of aurantio-obtusin for the treatment of inflammatory-related diseases. Since many anti-inflammatory actions have been attributed to the inhibition of prostaglandin synthesis, PGE_2_ was considered as one of the strongest inflammatory mediators in inflammation [42]. PGE_2_ is transformed from arachidonic acid via the COX-2 catalytic reaction, which may be affected by iNOS [42]. Aurantio-obtusin was shown to significantly suppress PGE_2_ production, and inhibit the COX-2 mRNA expression and protein expression in a dose dependent manner. These results attributed to the inhibitory effect of aurantio-obtusin on PGE_2_ production via blocking COX-2 gene and protein expression [43].

The accumulated evidence haa confirmed that interference NF-κB for studying the suppression of LPS-stimulated inflammatory cytokines (i.e., the encoding cytokines, IL-6 and TNF-α, and the inflammation-related enzymes, COX-2 and iNOS) [44,45]. Through investigating the specific mechanism whereby aurantio-obtusin inhibit the activation of NF-κB the phosphorylation of IκB and IKK, and protein expressions of NF-κB p65, three vital events of NF-κB to the nucleus was inhibited by aurantio-obtusin, in a concentration-dependent manner. In addition, we saw that the degradation and phosphorylation of IκB-α were also inhibited, thus, indicating that aurantio-obtusin may inhibit NF-κB activation by suppressing the phosphorylation of IκBα and IκBα, in LPS-induced RAW264.7 cells. The phosphorylation of the IκB protein is a vital step in the activation of NF-κB and is regulated by IκB kinase (IKKs). The IKK activity is also induced by activators of the NF-κB pathways [46,47]. In the present study, we observed that aurantio-obtusin suppressed the activation of IKK-α and IKK-β; this underlies its inhibitory activity on NF-κB activation.

## 4. Materials and Methods

### 4.1. Materials and Reagents

Aurantio-obtusin (98.3% purity) was purchased from Chinese Materials Research Center (Beijing, China). Dulbecco’s modified Eagle’s medium-high glucose (DMEM) and fetal bovine serum (FBS) were purchased from Biological Industries (Shanghai, China). Bacterial LPS (Escherichia coli serotype 026:B6, L8274), Dexamethasone and 2-(4,5-dimethyl- thiazol-2-yl)-2,5-diphenyltetrazolium bromide (MTT) were obtained from Sigma-Aldrich Co. (St. Louis, MO, USA). Griess reagent system, Cy3-labeled Goat Anti-Rabbit IgG, and DAPI were obtained from Beyotime (Beijing, China). IL-6 Mouse ELISA Kit and TNF-α Mouse ELISA Kit were obtained from eBioscience (San Diego, CA, USA). The Prostaglandin E2 Parameter Assay Kit was obtained from R&D Systems Inc. (Minneapolis, MN, USA). COX-2 In-Cell ELISA Colorimetric Detection kit was purchased from ThermoFisher Scientific (Waltham, MA, USA), and the mouse COX-2 antibody was obtained from BD Pharmingen (San Diego, CA, USA). Trizol reagent was obtained from Invitrogen (Paisely, UK). The PrimeScript II 1st Strand cDNA Synthesis Kit and SYBR premix were purchased from Takara Bio Inc. (Tokyo, Japan). The BCA Protein Assay Kit was obtained from Pierce (Waltham, MA, USA). Thermo Fisher Scientific provided the NE-PER Nuclear and Cytoplasmic Extraction Kit. Cell Signaling Technology (Beverly, MA, USA) provided the following antibodies: IKKβ, p-IKKα/β, NF-κB p65, IκBα, p-IκBα, Anti-rabbit IgG, and β-actin.

### 4.2. Cell Culture

RAW264.7 macrophages were obtained from Cell Culture Center of Chinese Academy of Medical Sciences (Beijing, China). Cells were cultured in DMEM supplemented with 10% FBS, 100 units/mL penicillin and 100 μg/mL streptomycin (Invitrogen, Carlsbad, CA, USA) at 37 °C, in a fully humidified incubator containing 5% CO_2_. For all experiments, cells were grown between 80%–90% confluence, and were subjected to no more than ten cell passages.

### 4.3. Cell Viability Assay

Cell viability was assessed using an MTT assay. RAW264.7 cells were seeded in a 96-well plate (5 × 10^4^ per well) and incubated overnight at 37 °C. The RAW264.7 cells were then treated with various concentrations of aurantio-obtusin (6.25–100 μM). After an additional 24 h of incubation at 37 °C, 10 μL of MTT (stock solution of 5 mg/mL) was added to arrive at a final concentration of 0.5 mg/mL in each well; additional incubation followed. The medium was discarded after 4 h and 100 μL DMSO added to every well to dissolve the formazan precipitate. Absorption at a wavelength of 570 nm was used to measure the using a multifunctional microplate reader (FlexStation 3, Molecular Devices, Silicon Valley, CA, USA).

### 4.4. Nitrite Assay

RAW264.7 macrophages were cultured in 96-well plates at a density of 1 × 10^4^ cells/well, containing 100 μL DMEM and 10% FBS; incubation was performed for 24 h. The cells were then pre-treated with various concentrations of aurantio-obtusin (6.25–50 μM) for 2 h, followed by stimulation with 0.2 μg/mL LPS for 24 h at 37 °C, in a 5% CO_2_ atmosphere. The Griess assay was used to measure the NO concentration in the medium. Fifty μL of the cell culture supernatant was added to an equal volume of Griess reagent (0.1% naphthylethylenediamine dihydrochloride and 1% sulphanilamide in 5% phosphoric acid) in a 96-well plate for mins, and the 540 nm (OD_540_) absorbance was detected. The quantity of nitrite was determined using a sodium nitrite standard curve. The results are presented as mean ± standard deviation (SD) of 6 replicates per representative experiment. 4-amino-5-methylamino-2′,7′-difluorescein (DAF-FM) which enhanced the intensity of fluorescence reacting with NO, on a laser scanning microscopy (LEICA, Frankfurt, Germany). RAW264.7 cells were seeded in 35-mm dishes at a density of 3 × 10^4^ cells for 24 h at 37 °C and 5% CO_2_. The cells were pretreated with aurantio-obtusin (50 μM) for 2 h, then stimulated with 0.2 μg/mL LPS for 24 h. Discarded the culture medium, then, 10 μM DAF-FM diacetate was incubated for 30 min and washed with PBS for three times. Digital images were generated with a LEICA laser scanning microscope with an excitation wavelength of 495 nm and an emission wavelength of 515 nm.

### 4.5. Determination of IL-6, TNF-α, COX-2 and PGE2 Production

RAW264.7 cells were cultured in 24-well plates (at a density of 2 × 10^4^ cells per well) and incubated for 24 h. Before stimulated with 0.2 μg/mL LPS for 24 h, the cells were treated with aurantio-obtusin (6.25–50 μM) for 2 h. The production of IL-6, TNF-α, COX-2 and PGE_2_ in the supernatant was determined using commercially ELISA Kits according to the manufacturer’s instructions [20,37,48,49].

### 4.6. Quantitative Real-Time PCR Analysis

Cells were cultured in 6-well plates (at a density of 10 × 10^5^ cells/well) for 24 h. The cells were pre-treated with various concentrations of aurantio-obtusin (6.25–50 μM) for 2 h, followed by stimulation with LPS (0.2 μg/mL) for 24 h. Total RNA was extracted using the TRIZOL reagent according to the manufacturer’s instruction. The PrimeScript II 1st Strand cDNA Synthesis Kit was used for reverse transcription according to the manufacture’s protocol. Real-time PCR detection system (MA-Smart, Suzhou, China) and SYBR were used for RT-PCR amplification of iNOS, IL-6, TNF-α, COX-2 and β-actin using the following conditions: 95 °C for 30 s, and 40 cycles of 95 °C for 5 s, and 60 °C for 34 s. A melt curve analysis was performed to verify the accuracy of the amplification at the end of each experiment. The primer sequences are listed in Table 1. Samples were compared using the relative C_T_ method. The fold increase or decrease was dependent on a blank control after the elimination of the housekeeping gene according to 2^−∆∆C_T_^ [50,51].

### 4.7. Western Blot Analysis

RAW264.7 cells were seeded in 6-well plates (10 × 10^5^ cells/well) for 24 h. The cells were pretreated with or without aurantio-obtusin (6.25–50 μM) for 2 h, followed by exposure to 0.2 μg/mL LPS for 12 h. The supernatants were discarded, and the cells washed with cold PBS (pH 7.4). The cells were then resuspended in lysis buffer (50 mM HEPES pH 7.0, 250 mM NaCl, 5 mM EDTA, 0.1% Nonidet P-40, 1 mM PMSF, 0.5 mM DTT, 5 mM NaF and 0.5 mM sodium orthovanadate) containing 5 mg/L leupeptin and aprotinin, and incubated for 20 min at 4 °C. The cell debris was removed by centrifugation and the supernatant collected. The protein concentrations of the cell lysates were detected by the BCA method (Pierce, Waltham, MA, USA). The cytosol and nuclear fractions were extracted using an NE-PER Nuclear and Cytoplasmic Extraction Kit (Thermo Fisher Scientific, San Jose, CA, USA) according to the manufacture’s protocol. Each cell lysate was loaded onto an 8% or 10% sodium dodecyl sulfate (SDS) polyacrylamide gel, followed by transfer of proteins to the PVDF membranes. After blocking the membrane in 0.01 % Tween 20 (TBST) containing 5% skimmed milk powder for 1 h, the membrane was incubated with the appropriate primary antibody (targeting IKKβ, p-IKKα/β, NF-κB p65, IκBα, p-IκBα, and β-actin; Cell Signaling Technology, MA, USA) at 1:1000 dilution overnight at 4 °C. The membrane was washed in TBST and incubated with the Anti-rabbit IgG secondary antibody (Cell Signaling Technology) at a 1:1000 dilution for 1 h. After three rounds of membrane washing in TBST, the immunoreactive bands were visualized and quantified using Gel Image system ver. 4.00 (Tanon, Shanxi, China).

### 4.8. Immunofluorescence Assay

RAW264.7 cells were seeded onto a 4-well chamber slide at a density of 5 × 10^3^ cells/mL. The cells were stimulated with LPS and then incubated with indicated concentrations of aurantio-obtusin. The cells then washed with PBS and fixed in 4.0% formaldehyde for 10 min at room temperature. Afterwards, the cells were permeabilized with 0.1% Triton X-100 for 4 min and washed twice with PBS. Then, the cells were incubated with anti-NF-κB p65 antibodies (1:100 with 2% BSA) overnight at 4 °C, followed by incubation with Cy3-labeled Goat Anti-Rabbit IgG for 1h at room temperature in the dark. Finally, the cells were washed with PBS and incubated with DAPI mounting medium for 10 min. The samples were then analyzed through fluorescence microscopy [52].

### 4.9. Statistical Analysis

Data represented as mean ± S.E.M with at least three independent experiments conducted. One-way ANOVA and two-tailed Student’s *t*-test were performed. *p* values < 0.05 were considered statistically significant. All statistical tests were carried out using the GraphPad Prism software (5.0, GraphPad Inc., San Diego, CA, USA).

## 5. Conclusions

In conclusion, the present study showed the potential of aurantio-obtusin as an anti-inflammatory agent. For the RAW264.7 macrophages that had been activated by LPS, aurantio-obtusin suppressed the LPS-induced production of TNF-α, IL-6, iNOS, COX-2 and PGE_2_, by inhibiting the activation of the NF-κB pathway. These results suggest that aurantio-obtusin may be a therapeutic agent that can be considered for further development of treatments for a variety of inflammation-related diseases. In addition, aurantio-obtusin may provide a new interpretation of the function of *Semen Cassiae* in inflammation. Further detailed and comprehensive studies on the anti-inflammatory effects of aurantio-obtusin in both in vitro and in vivo models of inflammatory diseases, are required to evaluate the therapeutic potential of aurantio-obtusin.

## Figures and Tables

**Figure 1 molecules-23-03093-f001:**
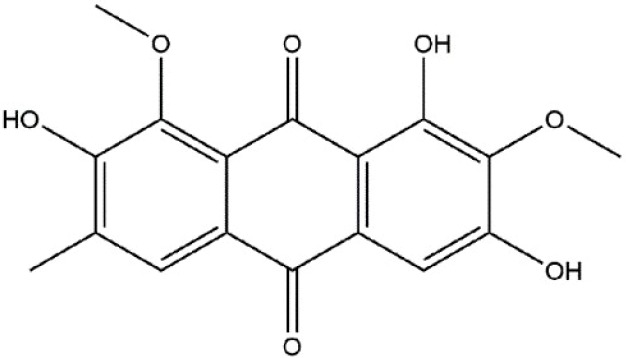
The chemical structure of aurantio-obtusin.

**Figure 2 molecules-23-03093-f002:**
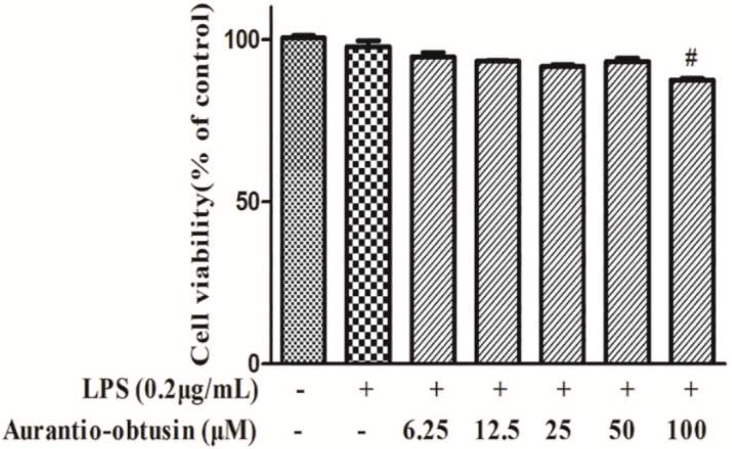
Effects of aurantio-obtusin on the cell viability of RAW264.7 cells. RAW264.7 cells incubated with various concentrations of aurantio-obtusin (6.25–100 μM) for 24 h followed by 0.2 μg/mL LPS stimulation. Cell viability was then measured using an MTT assay. ^#^
*p* < 0.05 represent significance when compared to the control cells.

**Figure 3 molecules-23-03093-f003:**
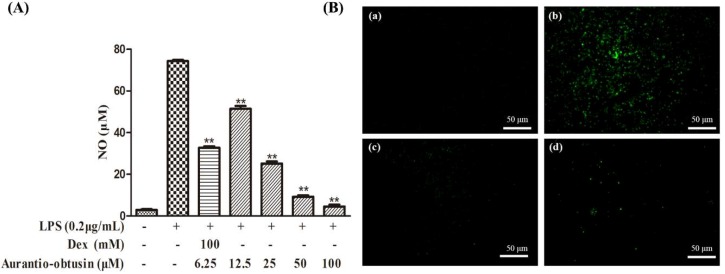
Effect of aurantio-obtusin on nitric oxide (NO) production in lipopolysaccharide (LPS)-activated RAW264.7 cells. (**A**) The RAW264.7 cells were treated with various concentrations (6.25–50 μM) of aurantio-obtusin for 2 h and then incubated with LPS (0.2 μg/mL) for 24 h. Data are presented as mean ± SD (*n* = 6). ** *p* < 0.01 represent significance when compared to LPS-only treated cells. (**B**) RAW264.7 cells were incubated with the indicated concentrations of aurantio-obtusin and 0.2 μg/mL LPS for 24 h. Intracellular NO production was evaluated with DAF-FM diacetate by laser scanning microscopy; (a) control (cells alone); (b) cells stimulated with LPS; (c) aurantio-obtusin (50 μM) was added under the condition of part (b); (d) 100 μM dexamethasone was added under the condition of part (b).

**Figure 4 molecules-23-03093-f004:**
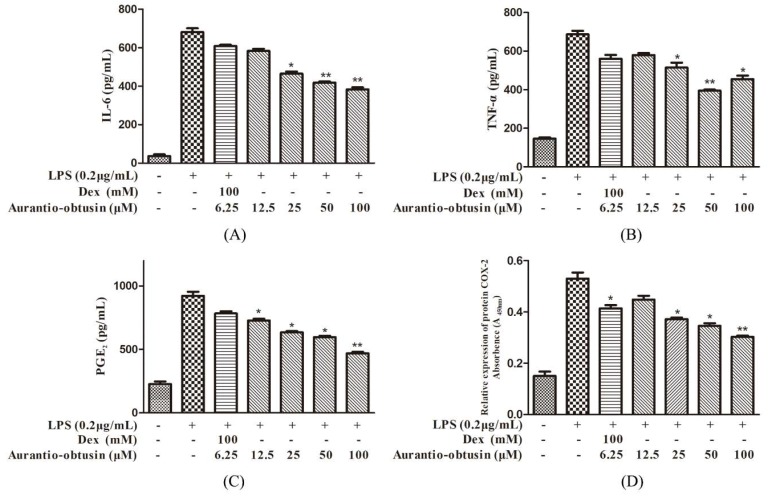
Effect of aurantio-obtusin on IL-6, TNF-α, PGE_2_ production and COX-2 protein expression in LPS-treated RAW264.7 cells. The cells were treated with various concentrations of aurantio-obtusin (6.25–50 μM) for 2 h, followed by stimulation with LPS (0.2 μg/mL) for 24 h. IL-6 (**A**), TNF-α (**B**) and PGE_2_ (**C**) production in the culture medium was determined by ELISA, and the COX-2 protein expression (**D**) analyzed by cell-based ELISA. The data are presented as mean ± SD (*n* = 3). * *p* < 0.05 and ** *p* < 0.01 represent significance when compared to LPS-only treated cells.

**Figure 5 molecules-23-03093-f005:**
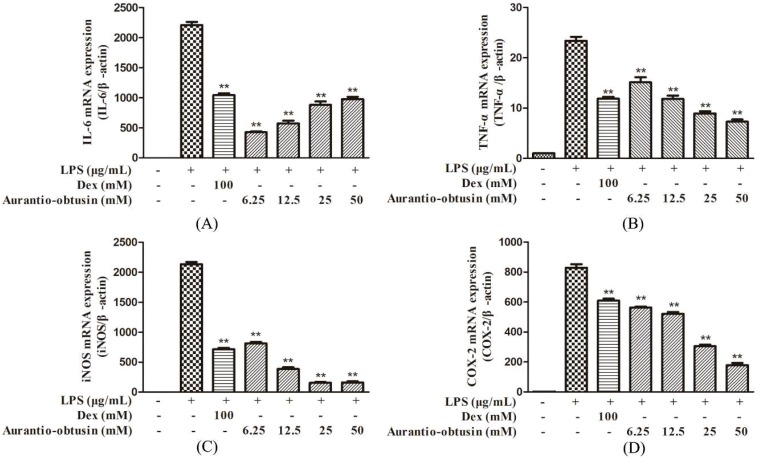
Effect of aurantio-obtusin on LPS-stimulated mRNA expression of IL-6, TNF-α, iNOS, and COX-2. RAW264.7 cells were pre-incubated with various concentrations of aurantio-obtusin (6.25–50 μM) for 2 h followed by stimulation with LPS (0.2 μg/mL) for 24 h. The mRNA expression ofIL-6 (**A**), TNF-α (**B**), iNOS (**C**) and COX-2 (**D**) was analyzed using Real-time RT-PCR. The data are presented as mean ± SDs (*n* = 3), ***p* < 0.01 represent significance when compared to LPS-only treated cells.

**Figure 6 molecules-23-03093-f006:**
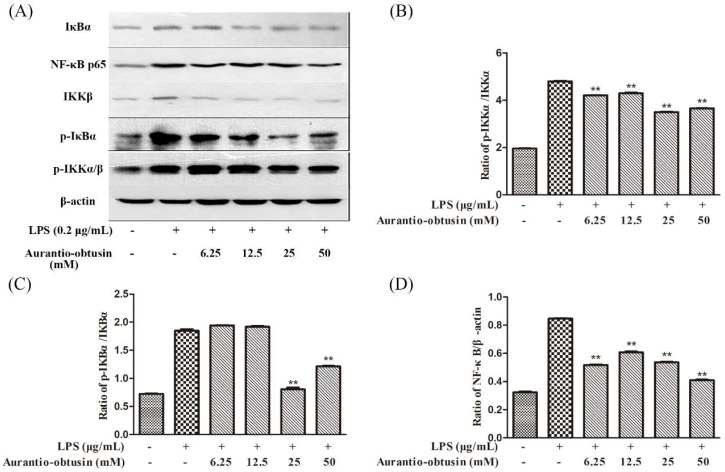
Effects of aurantio-obtusin on the expression of proteins associated with inhibition of NF-κB in LPS-stimulated RAW264.7 mouse macrophages. RAW264.7 cells were pre-incubated with various concentrations (6.25–50 μM) of aurantio-obstusin for 2 h, and stimulated with LPS (0.2 μg/mL) for 12 h. The β-actin protein was used as the internal control. (**A**) Expression of proteins associated with the inhibition of NF-κB was detected by a western blot analysis; Comparison of the levels of phosphorylated protein relative to the levels of their non-phosphorylated counterparts in the grey scale: p-IKKα/ IKKα (**B**) and p-IKBα/ IKBα (**C**); (**D**) Comparison of the levels of p65 relative to the level of their actin counterparts in the grey scale. ***p* < 0.01 represent significance when compared to LPS-only treated cells.

**Figure 7 molecules-23-03093-f007:**
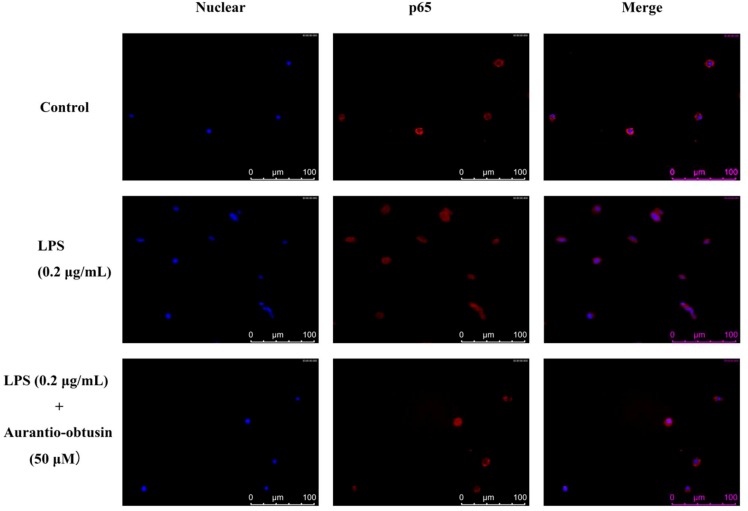
Nuclear localization of the p65 subunit of NF-κB was analyzed through immunofluorescence. Briefly, cells were fixed with 4.0% formaldehyde and permeabilized by incubating them with 0.1% Triton X-100. The cells were incubated with anti-NF-κB p65 antibodies overnight at 4 °C, and then with Cy3-labeled Goat Anti-Rabbit IgG for 1h at room temperature in the dark. Next, the cells were incubated with DAPI mounting medium for 10 min and the images were captured under a microscope. Fluorescent images of the cytoplasmic and nuclear fractions were merged to locate p65. Aurantio-obtusin inhibited the trans-location of p65 into the nucleus.

**Table 1 molecules-23-03093-t001:** Primers used for the quantitative real-time PCR.

Gene	Primer	Sequence (5′-3′)	PCR Product (bp)
β-Actin	Forward	TGTTACCAACTGGGACGACA	165
(NM_007393.3)	Reverse	GGGGTGTTGAAGGTCTCAAA	
iNOS	Forward	CACCTTGGAGTTCACCCAGT	170
(NM_010927.3)	Reverse	ACCACTCGTACTTGGGATGC	
COX-2	Forward	TGAGTACCGCAAACGCTTCTC	151
(NM_031168.1)	Reverse	TGGACGAGGTTTTTCCACCAG	
TNF-α	Forward	TAGCCAGGAGGGAGAACAGA	127
(NM_013693.2)	Reverse	TTTTCTGGAGGGAGATGTGG

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
