# Peer review of "Anti-Inflammatory Effects of Aurantio-Obtusin from Seed of Cassia obtusifolia L. through Modulation of the NF-κB Pathway"

_molecules, 2018, doi:10.3390/molecules23123093_

Round 1

Reviewer 1 Report

In the original paper: “Anti-inflammatory effects of Aurantio-obstusin from seed of Cassia obtusifolia L. through Modulation of the NF-κB pathway”, the anti-inflammatory activity on (LPS)-induced RAW264.7 cells of Aurantio-obtusin form seed of C. obtusifolia was described. In my opinion the manuscript is written carefully and it provides interesting results, however, I have some suggestions that will help in improving the work.

   Figure 1A and the Figure 1B should be presented as the separate figures (Figure 1 – the chemical structure and Figure 2 – effect on the cell viability).

  Abstract must contains the results of the experiments.

   The information that Aurantio-obtusin is an anthrquinone should be introduced to the Abstract. The species of plant producing the examined compound should be also added to the Abstract.

 It would be better if the keywords were not included in the title (anti-inflammation; Aurantio-obstusin; RAW264.7; NF-κB pathway). The keywords will assist in cross indexing the article.

  Regarding the introduction it could be improved (a few, more precise information on biological activity of Aurantio-obtusin is neded).

  Some words in the paper should be written in italic (e.g. Semen Cassiae in abstract).

  The abbreviation "IκB" and „IKK” are not explained in the text.

 The text should be proofread since there are some editorial errors (e.g. CO2 – subscripts are not included; 2h – too little spaces).

  It is important to check all the manuscript because of the wrong name „Aurantio-obstusin” used in some places (e.g. in the title of manuscript – I belive it should be: aurantio-obtusin).

Author Response

Dear reviewer:

Thank you very much for your letter with regard to our manuscript (Molecules-392743) together with the comments. Your suggestions are very valuable in improving the quality of our manuscript. Based on the comments and suggestions you proposed, we have read and understood your suggestions, made extensive modification on the manuscript. All corrections were marked clearly highlighted, please see the manuscript (marked in red).  

   Figure 1A and the Figure 1B should be presented as the separate figures (Figure 1 – the chemical structure and Figure 2 – effect on the cell viability).

AnswerThank you for your suggestion. We have changed the Figure 1B to Figure 2. Please see the Line 91-98.

  Abstract must contains the results of the experiments.

Answer : Thank you for your suggestion. The results of experiments have been added to the Abstract. Please see the Lines 25-31.   

   The information that Aurantio-obtusin is an anthrquinone should be introduced to the Abstract. The species of plant producing the examined compound should be also added to the Abstract.

Answer: Thank you for your suggestion. The detailed information of Aurantio-obtusin was added in the Abstract part. Please see the Lines 16-17.   

 It would be better if the keywords were not included in the title (anti-inflammation; Aurantio-obstusin; RAW264.7; NF-κB pathway). The keywords will assist in cross indexing the article.

Answer: Thank you for your suggestion. We have modified all keywords, please see the Line 35.

  Regarding the introduction it could be improved (a few, more precise information on biological activity of Aurantio-obtusin is needed).

Answer: Thank you for your suggestion. We have added more information on biological activity of Aurantio-obtusin. Please see the Line 71.

  Some words in the paper should be written in italic (e.g. Semen Cassiae in abstract).

Answer: Thank you for your suggestion. We have written some words in italic, please see the Lines 16-18, 78-79, and 215.  

  The abbreviation "IκB" and „IKK” are not explained in the text.

Answer: Thank you for your suggestion. We have added the relevant information with respect with "IκB" and IKK. Please see the Line 171 and 178.

 The text should be proofread since there are some editorial errors (e.g. CO2 – subscripts are not included; 2h – too little spaces).

Answer: Thank you for your suggestion. We have modified some editorial errors, please see the Lines through the whole manuscript.

  It is important to check all the manuscript because of the wrong name „Aurantio-obstusin” used in some places (e.g. in the title of manuscript – I belive it should be: aurantio-obtusin).

Answer: Thank you for your suggestion. We have checked and modified the Aurantio-obtusin. Please see the Lines through the whole manuscript.

Thanks for your time and consideration. I look forward to hearing from you.

Sincerely,

Jingyi Hou

Reviewer 2 Report

This manuscript describes the anti-inflammatory mechanism of Aurantio-obstusin on RAW264.7 cells. The anti-inflammatory effect of Aurantio-obstusin is associated with the activation of the nuclear factor -κB (NF-κB) pathway, especially the suppression of IκB and IKK phosphorylation, and the protein expression of NF-κB p65. The research design is appropriate, the results appear sound, but a few issues regarding the interpretation of the data and experimental protocols will need to be addressed before this manuscript can be published.

Major comments

1.    In Materials and Reagents, the tested compound, Aurantio-obtusin was purchased from Chinese Materials Research Center. Are there any documents providing the purity of the tested compound or any purification protocols can be shown in the part of materials and reagents? Purification and identification of isolated compound are very important in current study.  

Minor comments

1.      In Abstract, Line25, the (p value) can be showed as p < 0.01. Because the results in Fig 4 showed significant difference in all group with p < 0.01.

2.      Fig 1B2A345: The labels below the figures are too small to read. Please redraw the figure labels and make it larger.

3.      Table 1 can be moved to supplementary file.

Author Response

Dear reviewer:

Thank you very much for your letter with regard to our manuscript (Molecules-392743) together with the comments. Your suggestions are very valuable in improving the quality of our manuscript. Based on the comments and suggestions you proposed, we have read and understood your suggestions, made extensive modification on the manuscript. All corrections were marked clearly highlighted, please see the manuscript (marked in red).   

Major comments

1. In Materials and Reagents, the tested compound, Aurantio-obtusin was purchased from Chinese Materials Research Center. Are there any documents providing the purity of the tested compound or any purification protocols can be shown in the part of materials and reagents? Purification and identification of isolated compound are very important in current study.  

Answer: Thank you for your suggestion. The purity of the tested compound is 98.3% and added the purity of the tested compound in the manuscript (See the Line 264). The document is attached.  

Minor comments

1. In Abstract, Line25, the (p value) can be showed as p < 0.01. Because the results in Fig 4 showed significant difference in all group with p < 0.01.

Answer: Thank you for your suggestion. We have made corresponding modification with p value, please see the Line 29

2. Fig 1B2A345: The labels below the figures are too small to read. Please redraw the figure labels and make it larger.

Answer: Thank you for your suggestion. We have reedited the labels below the figures to make it larger. Please see the Figure 2- Figure 6.

3.       Table 1 can be moved to supplementary file.

Answer: Thank you for your suggestion. We have moved the Table 1 to supplementary file, please see the Line 330 and Supplementary Table 1.

Thanks for your time and consideration. I look forward to hearing from you.

Sincerely,

Jingyi Hou 

Reviewer 3 Report

Summary

Jingyi Hou and colleagues present a study, in which they investigate anti-inflammatory effects of the plant derived compound 1,3,7-Trihydroxy-2,8-diMethoxy-6-Methyl-9,10-anthracenedione or Aurantio-obtusin in the mouse macrophage cell line RAW264.7.“Aurantio-obstusin”, as used frequently in the manuscript, is wrong spelling. Tolerable doses were determined in a cell viability assay. Anti-inflammatory effects were tested in RAW cell after treatment with a moderate dose of 200 ng/ml LPS. Dexamethasone served as a positive control. Gene expression was determined on protein and mRNA level. As marker for inflammation served TNF-α, IL-6, COX-2, iNOS and PEG2. Further gene expression of IκBα, NF-κB p65, IKKβ and respective phosphorylation was determined in a dose dependent manner. Additional, they show that Aurantio-obtusin impairs the production of nitric oxide in a dose dependent manner. Microscopically the impairment of nuclear translocation of NF-κB p65 by Aurantio-obtusin is displayed in Fig.6. Here the photos are almost black in the pdf, what must be corrected. In the discussion the authors conclude, Aurantio-obtusin interferes with the NF-κB pathway, what is reasonable. The suppression of the activation of IKKα and IKKβ is regarded as an indicator for inhibition of NF-κB activation. Positively spoken, the authors do not speculate, on which molecule(s) Aurantio-obtusin acts. Missing TLR4 as the receptor for LPS is not mentioned.

Requests of the reviewer

The substance you work on is Aurantio-obtusin. You write several times “Aurantio-obstusin”; make sure to correct this.

In the formula in Figure 1 A and oxygen is missing. Use the correct formula.

Figure 6 in this pdf version is just black with some blue dots. Talk to the editor to obtain a better quality. After image processing I could see, what the real results are. As it is, it is not acceptable.

Reword all sentences, which are marked as odd in the list below. They are not clear.

Recommendations of the reviewer

The manuscript is concise and the basic experiments indicate anti-inflammatory effects on the LPS TLR4 NF-κB pathway. Since the experiments were only done with mouse macrophages and the effect on human cells is not shown, the conclusion “Aurantio-obtusin may be a therapeutic agent” is speculative, but not wrong.

In the same journal, Molecules, a paper was published in 2014, in which it is shown that Aurantio-obtusin is an extraordinary strong agonist for aryl hydrocarbon receptor. (doi:10.3390/molecules19044956, Characterization of natural aryl hydrocarbon receptor agonists from cassia seed and rosemary). This may be discussed. Since there are several papers, which describe the link between AhR and NF-κB, this would explain one possible mechanism for observed anti-inflammatory effects. E.g. doi: 10.1074/jbc.M113.505578. Cross-talk between aryl hydrocarbon receptor and the inflammatory response: a role for nuclear factor-κB.

Additionally you might discuss your citation 32 (J. Pharmacol. Sci. 2015, 128, 108-115) in the context of “, J Immunol March 15, 2008, 180 (6) 4218-4226; DOI: https://doi.org/10.4049/jimmunol.180.6.4218”. But this is only a suggestion, not a requirement.

Many scientist work on Toll-like receptor 4 (TLR4). Do not miss the chance to link your paper to this community. A good TLR4 antagonistic substance or modulator is highly welcome. LPS is a standard TLR4 agonist, which a led to the activation of NF-κB in Raw cells. You will get more attention.

Mistakes

Line 2 typo Aurantio-obstusin must read Aurantio-obtusin

Line 16 typo Aurantio-obstusin must read Aurantio-obtusin

Line 18 typo Aurantio-obstusin must read Aurantio-obtusin

Line 19 typo Aurantio-obstusin must read Aurantio-obtusin

Line 26 typo Aurantio-obstusin must read Aurantio-obtusin

Line 27 typo Aurantio-obstusin must read Aurantio-obtusin

Lines 26-29 In addition, a mechanistic study has revealed that the anti-inflammatory effect of Aurantio-obstusin is associated with the activation of the nuclear factor -κB (NF-κB) pathway, especially the suppression of IκB and IKK phosphorylation, and the protein expression of NF-κB p65.

This sentence is odd. The fact that LPS stimulates TLR4 on RAW cells, what causes an activation of NF-κB is well known. Apparently, Aurantio-obtusin is not “associated with the activation of the nuclear factor -κB (NF-κB) pathway”, vice versa, it has an inhibitory effect on the NF-κB pathway.

Line 32 typo Aurantio-obstusin must read Aurantio-obtusin

Line 34 typo Aurantio-obstusin must read Aurantio-obtusin

Line 45 “mediators”, The authors should mention explicitly reactive oxygen and nitrogen species here, since the they later measure nitric oxide production.

Lines 46-53 LPS is the prototype of Toll-like receptor 4 agonist. Would be reasonable to mention TLR4 here. This makes it easier to find the paper in the literature.

57-59 NF-κB is a protein complex, which can consists of 5 proteins (https://en.wikipedia.org/wiki/NF-%CE%BAB). Just talk about NF-κB p50/p65 what is more precise.

Line 69 typo Aurantio-obstusin must read Aurantio-obtusin

Line 73 typo Aurantio-obstusin must read Aurantio-obtusin

Line 90 Figure 1 A in comparison to the structure formula in ChemSpider http://www.chemspider.com/Chemical-Structure.136568.html in the middle on the lower side of the molecule an oxygen is missing at the double bond.

Line 102-104 is discussion in the result part

Line 124-126 is discussion in the result part

Line 149 Why in brackets (after LPS stimulation), is it a relevant information, otherwise delete. You mention it again in the next sentence again.

Line 157-160 is discussion in the result part

Line 160 Avoid information in brackets. Either the information is relevant or not.

Line 170-172 is a conclusion in the result part

Line 190 typo Aurantio-obstusin must read Aurantio-obtusin

Line 195 Figure 6. The quality of the photo is bad. It must be replaced, since the relevant details are not visible.

Line 210 outer membrane should read outer cell membrane

Line 211-212 see above, the main receptor for LPS is TLR4.

Line 217-219 provide a citation

Line 236-237 Sounds odd “This suppression was highly stimulated by LPS” LPS induces the expression but does not stimulate suppression.

Line 247 COX-2 and PGEs are NF-κB target genes, see https://www.bu.edu/nf-kb/gene-resources/target-genes/

Line 248 this is odd, “that NF-κB is a major mechanism” NF-κB is not a mechanism! Do you mean interference with NF-κB activation?

Line 250-253 This sentence is odd. “whereby Aurantio-obtusin activates NF-κB”. Doesn’t Aurantio-obtusin inhibit the activation of NF-κB? Please make shorter and clearer sentences here.

Line 258-259 odd since doubled: “we observed that Aurantio-obtusin suppressed the activation of IKK-α and IKK-β by Aurantio-obtusin”, please reduce to one Aurantio-obtusin.

Line 271 Thermo USA, which city?

Line 271 BD Pharmingen (USA), which city?

Line 275 Pierce (USA), which city. Your sources should be handled consistent.

Line 282 CO2 the number should be subscripted

Line 286 (5×104 per well) number 4 must be in superscript

Lines 288 – 289 do not separate number from unit.

Line 294 1×104 cells/well 104 number 4 must be in superscript

Line 297 CO2 ditto

Lines 304-305 How many cells were seeded?

Line 305 CO2 ditto.

Line 311 2×104 cells/well 104 number 4 must be in superscript

Line 317 ditto superscript

Line 331 ditto superscript

Line 344 “NF-κB Pathway Sampler Kit #9936” 9936 alone is confusing

Line 350 ditto superscript

Author Response

Dear reviewer:

Thank you very much for your letter with regard to our manuscript (Molecules-392743) together with the comments. Your suggestions are very valuable in improving the quality of our manuscript. Based on the comments and suggestions you proposed, we have read and understood your suggestions, made extensive modification on the manuscript. All corrections were marked clearly highlighted, please see the manuscript (marked in red).   

Requests of the reviewer 

The substance you work on is Aurantio-obtusin. You write several times “Aurantio-obstusin”; make sure to correct this.

AnswerThank you for your suggestion. The Aurantio-obstusin have been modified through the whole manuscript.

In the formula in Figure 1 A and oxygen is missing. Use the correct formula.

Answer: Thank you for your suggestion. We have corrected formula, please see the Figure 1.

Figure 6 in this pdf version is just black with some blue dots. Talk to the editor to obtain a better quality. After image processing I could see, what the real results are. As it is, it is not acceptable.

Line 195 Figure 6. The quality of the photo is bad. It must be replaced, since the relevant details are not visible.

Answer: Thank you for your suggestion. We have replaced images in the Figure 7. Please see the Line 195.

Reword all sentences, which are marked as odd in the list below. They are not clear.

Answer: Thank you for your suggestion. We have reedited the sentences, which were marked in the list below.

Recommendations of the reviewer

The manuscript is concise and the basic experiments indicate anti-inflammatory effects on the LPS TLR4 NF-κB pathway. Since the experiments were only done with mouse macrophages and the effect on human cells is not shown, the conclusion “Aurantio-obtusin may be a therapeutic agent” is speculative, but not wrong.

In the same journal, Molecules, a paper was published in 2014, in which it is shown that Aurantio-obtusin is an extraordinary strong agonist for aryl hydrocarbon receptor. (doi:10.3390/molecules19044956, Characterization of natural aryl hydrocarbon receptor agonists from cassia seed and rosemary). This may be discussed. Since there are several papers, which describe the link between AhR and NF-κB, this would explain one possible mechanism for observed anti-inflammatory effects. E.g. doi: 10.1074/jbc.M113.505578. Cross-talk between aryl hydrocarbon receptor and the inflammatory response: a role for nuclear factor-κB. 

Additionally you might discuss your citation 32 (J. Pharmacol. Sci. 2015, 128, 108-115) in the context of “, J Immunol March 15, 2008, 180 (6) 4218-4226; DOI:https://doi.org/10.4049/jimmunol.180.6.4218”. But this is only a suggestion, not a requirement. 

Many scientist work on Toll-like receptor 4 (TLR4). Do not miss the chance to link your paper to this community. A good TLR4 antagonistic substance or modulator is highly welcome. LPS is a standard TLR4 agonist, which a led to the activation of NF-κB in Raw cells. You will get more attention.

Answer: Thank you for your suggestion. We have reedited the relevant information and added the relevant information about TLR4. Please see the Lines 51-62, and 211-212.

Mistakes

Line 2 typo Aurantio-obstusin must read Aurantio-obtusin

Line 16 typo Aurantio-obstusin must read Aurantio-obtusin

Line 18 typo Aurantio-obstusin must read Aurantio-obtusin

Line 19 typo Aurantio-obstusin must read Aurantio-obtusin

Line 26 typo Aurantio-obstusin must read Aurantio-obtusin

Line 27 typo Aurantio-obstusin must read Aurantio-obtusin

Line 32 typo Aurantio-obstusin must read Aurantio-obtusin

Line 34 typo Aurantio-obstusin must read Aurantio-obtusin

Line 69 typo Aurantio-obstusin must read Aurantio-obtusin

Line 73 typo Aurantio-obstusin must read Aurantio-obtusin

Line 190 typo Aurantio-obstusin must read Aurantio-obtusin

AnswerThank you for your suggestion. We have reedited the Aurantio-obtusin through the whole manuscript.

Lines 26-29 In addition, a mechanistic study has revealed that the anti-inflammatory effect of Aurantio-obstusin is associated with the activation of the nuclear factor -κB (NF-κB) pathway, especially the suppression of IκB and IKK phosphorylation, and the protein expression of NF-κB p65.

This sentence is odd. The fact that LPS stimulates TLR4 on RAW cells, what causes an activation of NF-κB is well known. Apparently, Aurantio-obtusin is not “associated with the activation of the nuclear factor -κB (NF-κB) pathway”, vice versa, it has an inhibitory effect on the NF-κB pathway.

Answer: Thank you for your suggestion. We have reedited the sentence in Abstract Part. Please see the Line 29-31.

Line 45 “mediators”, The authors should mention explicitly reactive oxygen and nitrogen species here, since the they later measure nitric oxide production. 

Answer: Thank you for your suggestion. We have added the detailed information. Please see the Line 46.

Lines 46-53 LPS is the prototype of Toll-like receptor 4 agonist. Would be reasonable to mention TLR4 here. This makes it easier to find the paper in the literature. 

Answer: Thank you for your suggestion. We have added the relevant information about Toll-like receptor 4. Please see the Lines 51-52.

Line 211-212 see above, the main receptor for LPS is TLR4.

Answer: Thank you for your suggestion. We have added the relevant information in the Discussion Part. Please see the Lines 211-212.

57-59 NF-κB is a protein complex, which can consists of 5 proteins (https://en.wikipedia.org/wiki/NF-%CE%BAB). Just talk about NF-κB p50/p65 what is more precise.

Answer: Thank you for your suggestion. We have modified the description of NF-κB p50/p65, please see the Lines 55-57.

Line 90 Figure 1 A in comparison to the structure formula in ChemSpiderhttp://www.chemspider.com/Chemical-Structure.136568.html in the middle on the lower side of the molecule an oxygen is missing at the double bond. 

Answer: Thank you for your suggestion. We have corrected the formula in Figure 1, please see the Lines 91-92.

Line 102-104 is discussion in the result part

Line 124-126 is discussion in the result part

Line 157-160 is discussion in the result part

Line 170-172 is a conclusion in the result part

Answer: Thank you for your suggestion. We have deleted the duplicate information through the discussion part and result part. Please see the Lines 107-108, 124-126, 157-160 and 170-172.

Line 160 Avoid information in brackets. Either the information is relevant or not. 

Answer: Thank you for your suggestion. We have deleted the irrelevant information. Please see the Line 160.

Line 149 Why in brackets (after LPS stimulation), is it a relevant information, otherwise delete. You mention it again in the next sentence again. 

Answer: Thank you for your suggestion. We have deleted the irrelevant information. Please see the Line 149.

Line 210 outer membrane should read outer cell membrane

Answer: Thank you for your suggestion. We have modified the outer cell membrane, please see the Line 209.

Line 217-219 provide a citation

Answer: Thank you for your suggestion. We have added a citation. Please see the Line 214.

Line 236-237 Sounds odd “This suppression was highly stimulated by LPS” LPS induces the expression but does not stimulate suppression. 

Answer: Thank you for your suggestion. We have reedited the sentence, please see the line 236.

Line 247 COX-2 and PGEs are NF-κB target genes, see https://www.bu.edu/nf-kb/gene-resources/target-genes/

Answer: Thank you for your suggestion. We have modified the relevant information, please see the Line 248.

Line 248 this is odd, “that NF-κB is a major mechanism” NF-κB is not a mechanism! Do you mean interference with NF-κB activa tion? 

Answer: Thank you for your suggestion. We have been reedited the sentence. Please see the Line 246.

Line 250-253 This sentence is odd. “whereby Aurantio-obtusin activates NF-κB”. Doesn’t Aurantio-obtusin inhibit the activation of NF-κB? Please make shorter and clearer sentences here. 

AnswerThank you for your suggestion. We have been reedited the sentence to make it clearer. Please see the Line 248-249.

Line 258-259 odd since doubled: “we observed that Aurantio-obtusin suppressed the activation of IKK-α and IKK-β by Aurantio-obtusin”, please reduce to one Aurantio-obtusin. 

Answer: Thank you for your suggestion. We have deleted the duplicate information. Please see the Line 256-257.

Line 271 Thermo USA, which city?

Line 271 BD Pharmingen (USA), which city? 

Line 275 Pierce (USA), which city. Your sources should be handled consistent. 

Answer: Thank you for your suggestion. We have added the information of city. The source of all reagents have been handled consistent. Please see the Line 267-272.

Lines 288 – 289 do not separate number from unit. 

Answer: Thank you for your suggestion. We have separated number from unit through the whole manuscript.

Line 282 CO2 the number should be subscripted

Line 286 (5×104 per well) number 4 must be in superscript

Line 294 1×104 cells/well 104 number 4 must be in superscript

Line 297 CO2 ditto

Line 305 CO2 ditto. 

Line 311 2×104 cells/well 104 number 4 must be in superscript

Line 317 ditto superscript

Line 331 ditto superscript

Line 350 ditto superscript

Answer: Thank you for your suggestions. We have carefully checked the ditto superscript and superscript, and modified the mistakes through the whole manuscript.

Lines 304-305 How many cells were seeded? 

AnswerThank you for suggestion. We have added the number of cells. Please see the Line 304-305.

Line 344 “NF-κB Pathway Sampler Kit #9936” 9936 alone is confusing

Answer: Thank you for your suggestion. We have deleted the confusing number 9936. Please see the Line 343-345.

Thanks for your time and consideration. I look forward to hearing from you.

Sincerely,

Jingyi Hou